# FEDAKD: FEDERATED EDGE-ASSISTED ANOMALY-AWARE KNOWLEDGE DISTILLATION FOR 5G INTRUSION DETECTION

## ABSTRACT

The rise of 5G networks has exponentially increased the complexity and volume of network traffic, thereby strengthening the challenges in ensuring robust intrusion detection. Federated Learning (FL) emerges as a promising paradigm for collaborative anomaly detection, enabling multiple distributed clients to train a shared model without exchanging raw data, thus preserving privacy. However, FL in 5G environments wrestles with class imbalance, heterogeneous anomaly distributions, and constrained computational resources at edge devices. To address these issues, we propose a novel Federated Edge-Assisted Anomaly-Aware Knowledge Distillation (FEDAKD) framework designed for 5G network intrusion detection. FEDAKD integrates anomaly-aware sampling, teacher-student transformer architectures, and advanced aggregation techniques such as FedProx to enhance model performance while minimizing computational overhead. We conduct extensive evaluations on a 5G-specific intrusion dataset, demonstrating that FEDAKD outperforms baseline methods, including centralized training, Federated Averaging, and non-transformer classifiers, achieving higher weighted F1 scores and more accurate detection of various attack types. The results of the experiment underscore FEDAKD's efficacy in delivering scalable, privacy-preserving, and high-performance intrusion detection in modern 5G networks.

## 1 INTRODUCTION

The fifth generation of wireless networks (5G) has revolutionized connectivity, enabling a multitude of applications ranging from autonomous vehicles to the Internet of Things (IoT) Sheikhi & Kostakos (2023). However, the complexity and scale of 5G networks have increased, which has increased vulnerabilities to various cyber threats Farooqui et al. (2021); Storck & Duarte-Figueiredo (2020). Intrusion detection systems (IDS) are critical for safeguarding these networks, but traditional centralized approaches often suffer due to privacy concerns and the massive volume of data generated at the network edge Man et al. (2021); Sheikhi et al. (2024). *Federated Learning* (FL) presents a practicable solution by allowing distributed clients to collaboratively train a global model without transmitting raw data, thereby preserving privacy and reducing network bandwidth usage McMahan et al. (2017); Farooqui et al. (2021). Despite its advantages, FL in 5G environments faces considerable challenges, including class imbalance, where benign traffic significantly dominates malicious instances and heterogeneous anomaly distributions across different network components. In addition, edge devices in 5G networks frequently perform under severe computational and energy limitations, which requires lightweight and efficient models Dai et al. (2020); Sheikhi & Kostakos (2024). *Knowledge Distillation* (KD) offers a mechanism to mitigate these issues by transferring knowledge from a large, complex *teacher model* to a smaller, more efficient *student model* Gou et al. (2021); Park et al. (2019). With the integration of KD with FL, it is possible to maintain high detection performance while reducing the computational load on edge devices. However, existing FL-KD frameworks often dismiss the complexities of class imbalance and the diverse nature of anomalies inherent to 5G networks. In this paper, we introduce *Federated Edge-Assisted Anomaly-Aware Knowledge Distillation* (FEDAKD), a framework designed to enhance intrusion detection in 5G networks. FEDAKD combines anomaly-aware sampling, teacher-student transformer architectures, and advanced federated aggregation techniques such as FedProx to address class imbalance

and model heterogeneity. We evaluated FEDAKD using a comprehensive 5G-specific intrusion dataset, demonstrating its superiority over traditional baselines and non-transformer classifiers in terms of weighted F1 scores and anomaly detection accuracy.

The key contributions of this paper are as follows:

- **Introduction of FEDAKD**: We propose Federated Edge-Assisted Anomaly-Aware Knowledge Distillation (FEDAKD), a novel framework that combines knowledge distillation with federated learning to improve network intrusion detection performance in decentralized environments.

- **Anomaly-Aware Sampling**: We implement an anomaly-aware sampling strategy to effectively address class imbalance in intrusion datasets, ensuring that rare attack classes are sufficiently represented during model training.

- **Comprehensive Comparative Analysis**: We perform a thorough comparison of FEDAKD with standard federated learning baselines (FedAvg, FedProx, FedDyn, FedMD and FedProx) and centralized classifiers (LSTM, RandomForest and LogisticRegression), demonstrating the superior performance of FEDAKD in terms of weighted F1 scores and overall classification accuracy.

- **Extensive Experimental Evaluation**: We provide detailed experimental results, including confusion matrices and loss trend analyses, to illustrate the effectiveness of FEDAKD and its components in handling class imbalance and data heterogeneity.

- **Deployment Insights**: We offer guidelines and insights for deploying federated knowledge distillation techniques in real-world network security applications, emphasizing privacy preservation, scalability, and robustness.

## 2 BACKGROUND

Recent work on anomaly detection in 5G networks leverages advanced machine learning to tackle cell outages, congestion, and cyber-attacks, yet class imbalance between benign and malicious traffic remains a core challenge Porambage et al. (2021). Traditional sampling can discard signal or induce overfitting, whereas anomaly-aware sampling selectively balances benign and attack instances without sacrificing data integrity. Federated learning (FL) enables collaborative IDS modeling with privacy preservation and improves generalization across heterogeneous segments, with FedAvg as a standard baseline Bagdasaryan et al. (2020); McMahan et al. (2017). Heterogeneous data and imbalance persist; methods like FedProx add a proximal term to stabilize training and enhance robustness Li et al. (2020). Knowledge distillation (KD) supports model compression and lowers communication by transferring distributions rather than full parameters, which suits constrained edge devices Wang & Yoon (2021); Gad et al. (2024). Integrating FL with KD is therefore a promising direction for 5G anomaly detection, aligning privacy, communication efficiency, and robustness.

## 3 PROPOSED METHOD

We present the **Federated Edge-Assisted Anomaly-Aware Knowledge Distillation** (FEDAKD) framework, which integrates anomaly-aware sampling, teacher-student transformer architectures, and federated aggregation methods to enhance intrusion detection in 5G networks. Figure 1 illustrates the FEDAKD architecture.

### 3.1 ANOMALY-AWARE SAMPLING

Class imbalance is a dominant issue in intrusion detection, where benign traffic is extensively outnumbered by malicious instances Liu et al. (2020); Bedi et al. (2021). To mitigate this, FEDAKD employs anomaly-aware sampling, which strategically selects a proportionate number of benign and abnormal instances. Specifically, the framework samples a higher fraction of anomalous data to ensure that the model sufficiently learns from minority classes without being dominated by benign traffic. This approach preserves the diversity of attack types and maintains a balanced training set

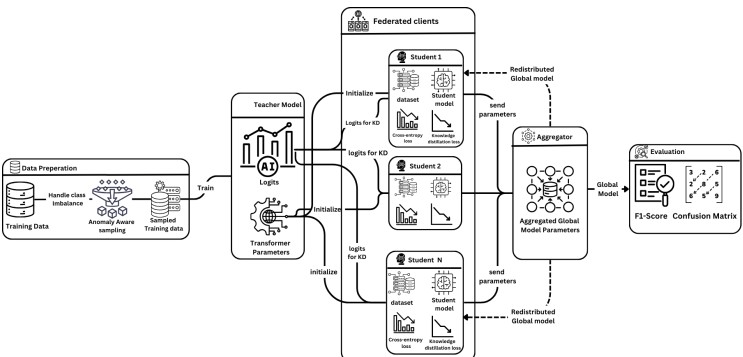

**Figure 1:** Overview of the FEDAKD architecture for 5G network intrusion detection.

across federated clients. The sampling process ensures that each federated client receives a representative subset of both benign and different anomaly classes, enhancing the model's ability to generalize across different attack types.

## 3.2 Teacher and Student Models

The teacher model is a transformer-based network leveraging the DistilBERT architecture. It is mainly trained on the sampled data set that is aware of anomalies, capturing comprehensive patterns and representations of the data. The architecture consists of a pre-trained DistilBERT model followed by a linear classification head to the number of intrusion classes. Each federated client hosts a lightweight student model, also based on DistilBERT but with a simplified classification head. The student models inherit the transformer layers' weights from the teacher, ensuring that they benefit from the teacher's pre-trained knowledge while remaining efficient.

## 3.3 Knowledge Distillation Loss

During training, the student models are optimized using a combination of **cross-entropy loss** for classification accuracy and **KL-divergence loss** to align the student's output distribution with that of the teacher. The total loss is defined as:

$$\mathcal{L} = \alpha \times \text{CE}(\hat{y}, y) + (1 - \alpha) \times \text{KL}\big(P_{\text{student}} \parallel P_{\text{teacher}}\big)$$

where $\alpha$ balances the two loss components, and a temperature parameter is applied to soften the teacher's output probabilities. In our implementation, $\alpha = 0.5$, and a temperature scaling factor of 10 is used to enhance the knowledge transfer effectiveness.

## 3.4 Federated Aggregation

We evaluate FedAvg federated aggregation strategy within FEDAKD to handle data heterogeneity and enhance model robustness. FedAvg is the standard federated averaging approach, which computes a weighted average of client model parameters based on client data sizes Mora et al. (2024). The aggregation process utilizes client performance metrics, such as weighted F1 scores, to weight the contributions of individual client models effectively. This ensures that clients with better performance have a more significant impact on the global model, enhancing overall detection capabilities.

## 3.5 FEDAKD Framework Workflow

The FEDAKD framework operates through five steps: **Data Preprocessing** to apply anomaly-aware sampling and balance the training dataset; **Model Initialization** to train the teacher model centrally and distribute its transformer parameters to all federated clients; **Federated Training** where, in each round, clients train their student models using knowledge distillation and local data; **Aggregation** to combine client models with the chosen federated aggregation strategy; **Evaluation** to assess the aggregated global model on the test dataset.

## 3.6 ALGORITHMIC OVERVIEW

The following outlines the high-level workflow of the FEDAKD framework:

1. Data Preparation: Start with a balanced dataset split into training and testing subsets. Apply anomaly-aware sampling to training data to address class imbalance.

2. Teacher Model Training: Train a central teacher model using the sampled training data.

3. Client Distribution: Distribute the teacher model's transformer parameters to all federated clients.

4. Local Training with Knowledge Distillation: Each client initializes its student model with the teacher's transformer parameters and trains it on local data using both cross-entropy and knowledge distillation losses.

5. Model Aggregation: After local training, client models are aggregated using FedAvg.

6. Global Model Update: Update the global model with the aggregated parameters.

7. Evaluation: After completing the federated rounds, evaluate the global model on the test dataset to assess performance.

## 4 ALGORITHM PSEUDOCODE

The pseudocode in Algorithm 1 succinctly outlines the main steps of FEDAKD framework. It begins with anomaly-aware data preprocessing and centralized teacher model training, followed by the initialization of client-side student models. During federated training, each client locally updates its model via knowledge distillation and subsequently contributes to a weighted aggregation of transformer parameters, leading to an updated global model that is finally evaluated on the test dataset.

---

**Algorithm 1** Federated Edge-Assisted Anomaly-Aware Knowledge Distillation (FEDAKD)

**Require:** Training dataset $D$, anomaly-aware sampling fractions $\gamma_{\text{anomaly}}$ and $\gamma_{\text{normal}}$,
    number of clients $K$, rounds $R$, teacher model $T$, student model $S$,
    KD hyper-parameter $\alpha$, temperature $T_{\text{temp}}$, and noise $\sigma$.
**Ensure:** Global student model $S_{\text{global}}$.

0: **Data Preprocessing**:
0: Apply anomaly-aware sampling on $D$ to create balanced training data $D_s$.
0: Split $D_s$ into training and test sets; partition $D_s$ among $K$ clients.
0: **Teacher Model Training**:
0: Train teacher model $T$ centrally on $D_s$.
0: **Client Initialization**:
0: **for** each client $k = 1, \ldots, K$ **do**
0:     Initialize student model $S_k$ with transformer's parameters from $T$.
0: **end for**
0: **Federated Training**:
0: **for** $r = 1$ to $R$ **do**
0:     **for** each client $k = 1, \ldots, K$ (in parallel) **do**
0:         **Local Update**: Train $S_k$ on local data with combined loss

$$\mathcal{L} = \alpha \, \text{CE}\big(S_k(x), y\big) + (1 - \alpha) \, T_{\text{temp}}^2 \, \text{KL}\Big(\text{softmax}\Big(\frac{S_k(x)}{T_{\text{temp}}}\Big) \,\Big\|\, \text{softmax}\Big(\frac{T(x)}{T_{\text{temp}}}\Big)\Big)$$

0:         Compute local performance metric $w_k$ (e.g., weighted F1-score).
0:     **end for**
0:     **Aggregation**:
0:     Collect updated transformer parameters $\theta_k$ from all clients.
0:     Aggregate as

$$\theta^{(r+1)} = \sum_{k=1}^{K} \left(\frac{w_k}{\sum_{j=1}^{K} w_j} \theta_k\right) + \mathcal{N}(0, \sigma^2)$$

0:     Update the global student model $S_{\text{global}}$ with $\theta^{(r+1)}$.
0: **end for**
0: **Global Evaluation**:
0: Evaluate $S_{\text{global}}$ on the test dataset. =0

---

# 5 EXPERIMENTAL SETUP

## 5.1 DATASET COLLECTION

The dataset was generated on a custom-built 5G network testbed integrating the Open5GS core with Dockerized services to simulate both Internet and IoT environments (Figure 2). We configured **network slicing** in the 5G core to isolate traffic streams and emulate real-world multi-service operation; **deployed Dockerized services** to represent heterogeneous network functions and produce realistic flows; **generated normal traffic** to establish a baseline of benign behavior; **simulated attack traffic** to capture the network's response to diverse cybersecurity threats; **continuously captured and processed data**, extracting features across benign and malicious flows; and **prepared the data for analysis** through transfer and preprocessing steps, enabling evaluation under both stable conditions and attack scenarios. This workflow yields a comprehensive and realistic dataset that captures benign and attack traffic across Internet and IoT slices.

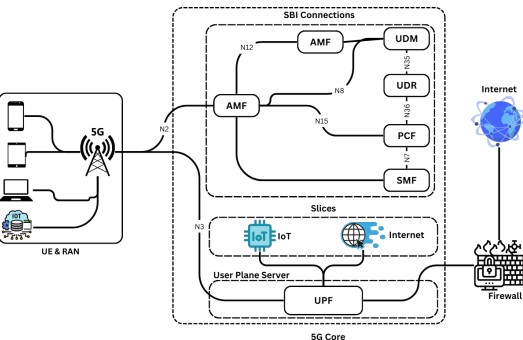

**Figure 2:** The structure of the 5G testbed.

This structured workflow ensures the creation of a comprehensive dataset that accurately represents network performance and security behavior in a modern 5G environment. The use of network slicing and Dockerized services enhances the realism and scalability of the simulation, which enhances the robustness of the dataset.

## 5.2 ATTACK SIMULATIONS

The dataset includes simulated attacks to evaluate the 5G core and services across both internet and IoT slices under realistic conditions. We model: **DDoS** on the internet slice to overwhelm services; **SQL Injection** against a web service to manipulate or access data; **Brute Force** on a login interface to gain unauthorized access; **MITM** on the internet slice to intercept and alter client–server traffic; **DoS** on the IoT MQTT broker to disrupt device communication; **Device Spoofing** to impersonate legitimate IoT devices; **Unauthorized Data Access** via vulnerability scans on the IoT slice; and **Eavesdropping** on IoT traffic to capture device communications. These simulations mirror real-world threats and yield a diverse dataset of benign and malicious behaviors for rigorous evaluation.

## 5.3 DATASET AND PREPROCESSING

**Data Volume and Sampling.** The original dataset contains 1,753,454 training samples and 194,829 testing samples, each with 29 features. To reduce computational overhead and address class imbalance, we apply an *anomaly-aware sampling* technique. This process draws fewer samples from the majority (*benign*) class and a proportionally larger share from each minority attack class, resulting in a *training subset* of 108717 rows and a *testing subset* of 12113 rows. As shown in Table 1, *DoS_MQTT* and *DDoS* account for the majority of attack samples, while infrequent classes such as *MITM* and *Unauthorized Data Access* remain comparatively rare.

**Federated Partitioning.** For the federated setup, the sampled training subset of 108,717 rows is divided among five clients, each receiving around 21,700 samples. The split is carefully designed to include representative proportions of minority classes (e.g., *Brute Force*, *MITM*) at every client,

**Table 1:** Class distributions in the anomaly-aware sampled subsets.

| Attack Class | Train Count | Test Count |
|---|---|---|
| Benign | 66,631 | 7,404 |
| DoS_MQTT | 25,052 | 2,720 |
| DDoS | 16,484 | 1,901 |
| Eavesdropping | 361 | 37 |
| MITM | 68 | 11 |
| SQL Injection | 54 | 10 |
| Unauthorized Data Access | 31 | 10 |
| Brute Force | 26 | 10 |
| Device Spoofing | 10 | 10 |

even if these attack categories appear in small numbers overall. This partitioning strategy reflects real-world scenarios where distributed nodes collect diverse subsets of benign and malicious traffic while preserving critical anomalies in every local dataset.

## 5.4 Implementation Details

All experiments are implemented in Python using `PyTorch` and `Transformers`. Key hyperparameters are: **Max Sequence Length** 128; **Batch Size** 16 for clients and 32 for centralized training; **Learning Rate** $2 \times 10^{-5}$; **Number of Clients** 5; **Federated Rounds** 5; **Epochs** 2 for centralized training and 1 per federated round. The implementation ensures that the computational and memory constraints of edge devices are respected, allowing the student models to operate efficiently without compromising detection performance.

## 5.5 Evaluation Metrics

We employ three complementary metrics to evaluate model performance: **Weighted F1 Score**, which accounts for class imbalance by weighting each class-wise F1 by its prevalence; the **Confusion Matrix**, which provides detailed insight into performance across classes; and the **Classification Report**, which includes precision, recall, and F1 scores for each class. These metrics collectively offer a comprehensive view of the model's ability to accurately detect and classify various intrusion types, especially in the presence of class imbalance.

## 5.6 Experimental Workflow

The experimental workflow proceeds as follows: **Data Preparation**, apply anomaly-aware sampling to the training data and split it among federated clients; **Teacher Model Training**, train the teacher model centrally on the sampled training data; **Federated Training**, conduct federated rounds where each client trains its student model via knowledge distillation; **Model Aggregation**, aggregate client models using the selected federated aggregation strategy; **Evaluation**, assess the aggregated global model on the test dataset and compare against baselines. This structured approach ensures that each component of the FEDAKD framework is systematically evaluated, providing clear insights into its effectiveness and areas for improvement.

# 6 Results and Discussion

In this section, we present and discuss the performance of our proposed FEDAKD method in comparison with three federated baselines (FedAvg and FedProx) and two centralized baselines (RandomForest and LogisticRegression).

## 6.1 Overall Performance

Table 2 and Figure 3a illustrate the weighted F1 scores for all methods on the test set. FEDAKD achieves a high weighted F1 score of **0.9950**, substantially outperforming FedAvg (**0.4960**) and other baseline federated methods such as FedProx, FedDyn, and FedMD, which all record near-zero weighted F1 scores. Among centralized models, LSTM attains an excellent weighted F1 of **0.9969**, while Random Forest and Logistic Regression follow closely at **0.9922** and **0.9904**, respectively. These results confirm that centralized models can indeed achieve outstanding performance when

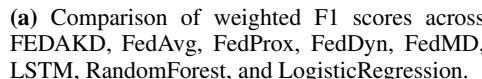

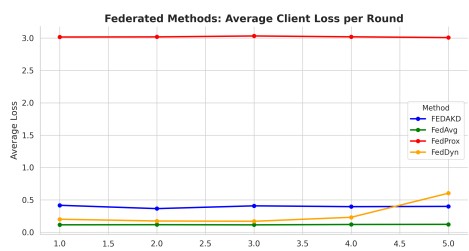

**(a)** Comparison of weighted F1 scores across FEDAKD, FedAvg, FedProx, FedDyn, FedMD, LSTM, RandomForest, and LogisticRegression.

**(b)** Average client loss per round for FEDAKD, FedAvg, FedProx, FedDyn, and FedMD.

**Figure 3:** Side-by-side metrics: (a) weighted F1 scores and (b) average client loss per round.

sufficient training data is available in a single location. Nevertheless, FEDAKD's near-centralized performance highlights the effectiveness of knowledge distillation within the federated setting.

**Table 2:** Weighted Classification Metrics for All Methods

| Method | Accuracy | Weighted Precision | Weighted Recall | Weighted F1 |
|---|---|---|---|---|
| FEDAKD | 0.9960 | 0.9939 | 0.9960 | 0.9950 |
| FedAvg | 0.6155 | 0.4673 | 0.6155 | 0.4960 |
| FedProx | 0.0009 | 0.0000 | 0.0009 | 0.0000 |
| FedDyn | 0.0010 | 0.0000 | 0.0010 | 0.0000 |
| FedMD | 0.0009 | 0.0042 | 0.0009 | 0.0002 |
| LSTM | 0.9972 | 0.9968 | 0.9972 | 0.9969 |
| RandomForest | 0.9942 | 0.9928 | 0.9942 | 0.9922 |
| LogisticRegression | 0.9931 | 0.9897 | 0.9931 | 0.9904 |

## 6.2 FEDERATED METHODS: CLIENT LOSS PER ROUND

Figure 3b illustrates the average client loss per round for FEDAKD, FedAvg, FedProx, and FedDyn. The FEDAKD curve (blue) remains fairly stable around **0.4** across the rounds, reflecting its combined loss of cross-entropy and Kullback–Leibler divergence for knowledge distillation. In contrast, FedAvg (green) exhibits the *lowest* training loss at roughly **0.1**, yet it fails to generalize well and produces only a **0.4960** weighted F1 score due to imbalanced data. FedProx (red) consistently stays around a **3.0** loss, suggesting that its proximal term heavily constrains updates without improving classification in this multi-class scenario. Finally, FedDyn (orange) starts near **0.2** but ends around **0.6** by the final round, indicating convergence difficulties when dealing with the highly imbalanced classes. These varying loss curves underscore that low training loss alone does not guarantee robust performance; methods like FEDAKD benefit from incorporating knowledge distillation to balance reduction in training loss with strong generalization.

## 6.3 ANALYSIS OF CONFUSION MATRICES

To gain deeper insight into class-specific predictions, we examine the confusion matrices of the primary methods. In particular, we focus on FEDAKD (best performer), FedAvg (federated baseline), Random Forest (best centralized), and Logistic Regression (centralized baseline). Figures 4a–4f present these results. Figure 4a shows that FEDAKD classifies the major attack types with impressive accuracy, including *DoS_MQTT* (2720/2720), *DDoS* (1900/1901), and *benign* (7403/7404). It also performs well on *Eavesdropping* (33/37). However, some low-frequency classes are entirely misclassified: for instance, all *MITM* samples (11) are predicted as *Brute Force*, and all *Unauthorized Data Access* samples (10) are split among other classes (e.g., *Eavesdropping*, *benign*). *Device Spoofing*, *Brute Force*, and *SQL Injection* also see zero correct predictions, with their instances scattered across multiple other labels. Despite these errors on rare attacks, the confusion matrix highlights FEDAKD's effectiveness in handling the dominant classes, a result of combining federated knowledge distillation and ensemble aggregation. While class imbalance remains challenging

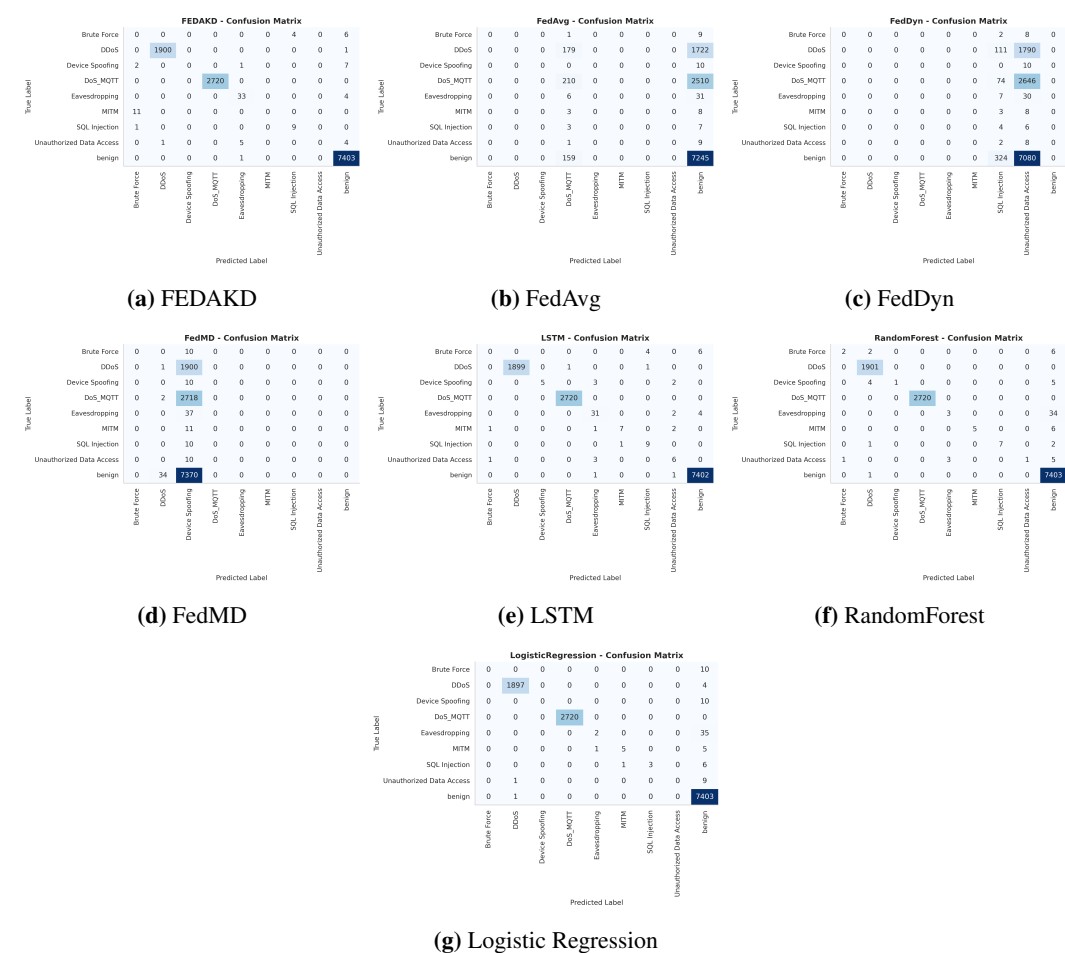

**(a)** FEDAKD        **(b)** FedAvg        **(c)** FedDyn

**(d)** FedMD        **(e)** LSTM        **(f)** RandomForest

**(g)** Logistic Regression

**Figure 4:** Confusion matrices for all evaluated models in a compact layout.

for minority categories, the approach still provides strong overall performance compared to other federated baselines.

As illustrated in Figure 4b, FedAvg largely collapses predictions into the *benign* class (7,245 out of 7,404 benign samples correctly identified), but it misclassifies the vast majority of non-benign examples. In particular, it fails to classify any *DDoS* attacks correctly (0 out of 1,901), instead predicting them mostly as *DoS_MQTT* (179) or *benign* (1,722). Although FedAvg does correctly label some *DoS_MQTT* samples (210 out of 2,720), other low-frequency categories such as *Device Spoofing* and *SQL Injection* are almost entirely subsumed by the *benign* label. This skewed prediction behavior explains FedAvg's modest weighted F1 score of **0.4960**, as it lacks mechanisms like knowledge distillation or class re-weighting to handle the highly imbalanced nature of this intrusion dataset. Figure 4c and the corresponding classification report reveal that FedDyn struggles severely with this imbalanced intrusion detection task. Overall accuracy is effectively **0.00**, and most attack categories show near-zero precision, recall, and F1 scores. The only exceptions are *SQL Injection* and *Unauthorized Data Access*, which achieve recalls of 0.40 and 0.80 respectively, though even these returns negligible F1 scores due to the model's failure to correctly classify other classes. In particular, high-frequency classes such as *benign*, *DoS_MQTT*, and *DDoS* are almost entirely misclassified, underscoring FedDyn's inability to learn useful decision boundaries under the highly imbalanced distribution. This indicates that, without additional regularization or data-balancing strategies, FedDyn collapses in the face of class imbalance and fails to provide meaningful detection performance. As shown in Figure 4d, FedMD exhibits a highly imbalanced prediction behavior, assigning nearly all inputs, regardless of their true class to the *Device Spoofing* category. For example, it predicts **1900** out of 1901 *DDoS* samples and **2718** out of 2720 *DoS_MQTT* samples as *Device Spoofing*,

along with misclassifying nearly every benign and low-frequency attack instance in the same way. It is significant that **7370** benign samples have been misclassified and incorrectly labeled as *Device Spoofing*, further highlighting the model's severe overfitting to a single class.

Figure 4e demonstrates that the LSTM model classifies the major classes with high accuracy, including *DDoS* (1899/1901), *DoS_MQTT* (2720/2720), and *benign* (7402/7408). However, minority classes such as *Brute Force*, *MITM*, and *Unauthorized Data Access* still pose a challenge. For instance, *Brute Force* samples are consistently misclassified, and some *MITM* traffic is confused with *benign* or other attacks. Despite these misclassifications on rare categories, LSTM's centralized training with ample data enables it to achieve near-perfect performance on the most frequent attack types. This underscores the limitations of purely data-driven sequence models in handling minority classes without additional balancing or regularization techniques. Among the centralized methods, RandomForest achieves one of the most robust weighted F1 scores of 0.9865 and demonstrates near-perfect classification for the dominant classes *DDoS* (1901/1901), *DoS_MQTT* (2720/2720), and *benign* (7403/7404). However, some minority classes exhibit considerable misclassifications. For instance, only 1/10 *Device Spoofing* samples are correctly detected, and *Eavesdropping* (3/37) is often labeled as *benign*. These confusion patterns reveal that while centralized training with complete data grants clear advantages, class imbalance still poses a challenge, even in single-site scenarios. Although Logistic Regression achieves an overall weighted F1 of 0.9842, the confusion matrix in Figure 4g shows that it struggles to correctly identify certain minority classes. For instance, all ten *Device Spoofing* samples are misclassified as *benign*, and *Eavesdropping* (2/37) is also frequently mislabeled. In contrast, the model performs nearly perfectly on the majority classes: *DoS_MQTT* (2720/2720) and *DDoS* (1897/1901) see minimal errors. Thus, while the centralized approach grants Logistic Regression a robust overall performance, issues with class imbalance remain evident for lower-frequency attacks.

### 6.4 DISCUSSION

**Effectiveness of KD in Federated Learning:** Integrating knowledge distillation into FL (FEDAKD) markedly improves performance: Student models learn from local data and the knowledge distilled from the teacher, producing a weighted F1 of **0.9950** and approaching centralized results in dominant classes while preserving data privacy. **Shortcomings of Vanilla FedAvg, FedProx, and FedDyn:** FedAvg, FedProx, and FedDyn converge on client loss but struggle with class imbalance; FedAvg reaches only **0.4960** weighted F1, and FedProx/FedDyn fall to near-zero on minority classes. Confusion matrices show frequent mislabeling of non-benign attacks as benign, highlighting the need for KD or class-aware reweighting to capture rare attacks. **Comparison with Centralized Methods:** Centralized LSTM, RandomForest, and LogisticRegression benefit from pooled data and excel on frequent classes, yet still falter on rare ones. FEDAKD nearly matches centralized performance on dominant classes without data pooling; rare categories such as *MITM* and *SQL Injection* remain challenging, but the quantitative gains and privacy benefits support practical deployment. **Implications for Intrusion Detection:** combining anomaly-aware sampling, FL, and KD forms a robust IDS approach: leveraging local and global signals delivers high overall accuracy despite imbalance. Future work should reduce errors on rare attacks via refined sampling, adaptive losses, and stronger distillation, advancing resilient, privacy-preserving intrusion detection.

## 7 CONCLUSION

We introduced the *Federated Edge-Assisted Anomaly-Aware Knowledge Distillation* (FEDAKD) framework, designed to enhance intrusion detection in 5G networks through the integration of anomaly-aware sampling, teacher-student transformer architectures, and advanced federated aggregation techniques. FEDAKD effectively addresses class imbalance and model heterogeneity, achieving superior weighted F1 scores compared to traditional baselines and non-transformer classifiers. Our comprehensive evaluations underscore FEDAKD's potential as a scalable, privacy-preserving solution for robust anomaly detection in modern 5G infrastructures. Future research will focus on integrating personalized federated learning algorithms, expanding the framework to handle real-time streaming data, and deploying FEDAKD in live 5G network environments to validate its efficacy and adaptability under dynamic conditions.

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
