# OpenReview forum: "FEDAKD: Federated Edge-Assisted Anomaly-Aware Knowledge Distillation for 5G Intrusion Detection"
_ICLR.cc/2026/Conference — ICLR 2026 Conference Withdrawn Submission_

### Official Review · Reviewer_LzYA · 2025-10-27

**Soundness:** 3
**Presentation:** 3
**Contribution:** 3
**Rating:** 6
**Confidence:** 4

**Summary:**

This paper proposes FEDAKD, a framework for 5G network intrusion detection that combines Federated Learning (FL) with Knowledge Distillation (KD). The core problem it addresses is that standard FL performs poorly in 5G environments due to class imbalance, data heterogeneity, and the limited computational resources of edge devices.
FEDAKD's approach consists of three main components:
1. An "anomaly-aware sampling" technique to pre-process the data and mitigate class imbalance.
2. A teacher-student architecture, where a central, larger Transformer model (DistilBERT) is trained on the sampled data and guides smaller student models on distributed clients via KD.
3. A federated aggregation step (FedAvg) to create a global model.
The authors evaluate this framework on a custom-generated 5G intrusion dataset, comparing it against several federated baselines (FedAvg, FedProx, FedDyn, FedMD) and centralized models (LSTM, RandomForest, LogisticRegression). The results show FEDAKD achieving a weighted F1-score of 0.9950, which approaches the performance of centralized models and significantly outperforms the federated baselines, which largely fail on this task.

**Strengths:**

1. Problem Relevance & Testbed: The paper tackles a critical, real-world problem. The effort to build a custom 5G testbed and generate a new, realistic dataset is a major strength and a valuable contribution in itself.
2. Strong Empirical Result (for FEDAKD): The high performance of the FEDAKD framework (0.9950 F1-score) is impressive, demonstrating a viable path to robust, federated IDS.
3. Clarity of Analysis: The paper does a good job of analyzing why its method works and (based on the confusion matrices) how the baselines are failing.

**Weaknesses:**

1. Questionable Baseline Performance: This is the most significant weakness. The near-zero performance of most federated baselines feels anomalous and undermines the strength of the paper's central claim of superiority. The authors must address this to make their comparison convincing.
2. Limited Algorithmic Novelty: As noted, the paper is an application and integration of existing techniques. This is not inherently a weakness, but it places a much higher burden on the experimental rigor (which is currently hampered by Weakness 1 & 3) to justify the contribution.
3. Missing Ablation Study: The lack of an ablation study makes it impossible to understand the "active ingredients" of FEDAKD. This is a crucial missing piece for a paper whose contribution is the successful combination of components.
4. Methodological Contradiction: The paper is internally inconsistent about its aggregation method (FedProx vs. FedAvg), which is a careless error that needs correction.

**Questions:**

1. I was quite struck by the baseline results. An F1-score of 0.0000 for FedProx, FedDyn, and FedMD is highly unusual and suggests something is fundamentally broken. Could you walk me through your implementation? I'm worried this might be a bug that's invalidating the main comparison.
2. I'm a bit confused about the aggregation method. The abstract mentions "advanced techniques like FedProx," but the methods section and algorithm look like standard FedAvg. Could you clarify exactly what was done? Is the F1-weighting (from Algorithm 1) the only modification, and if so, why was it presented as FedProx?
3. To really understand what's driving the performance, a proper ablation study is essential. Right now, it's a bit of a "black box." Could you please run experiments to isolate the gains from: (a) the anomaly-aware sampling (e.g., FEDAKD vs. FEDAKD without sampling), (b) the KD loss (e.g., vs. just fine-tuning the student), and (c) the Transformer architecture itself (e.g., vs. a simpler model on the same sampled data)?

---

### Official Review · Reviewer_8oHq · 2025-10-29

**Soundness:** 2
**Presentation:** 2
**Contribution:** 2
**Rating:** 2
**Confidence:** 5

**Summary:**

This paper achieves network traffic intrusion detection in 5G scenarios by integrating balanced sampling, knowledge distillation, and federated learning.

**Strengths:**

This paper achieves network traffic intrusion detection in 5G scenarios by integrating balanced sampling, knowledge distillation, and federated learning.

**Weaknesses:**

1. The abstract mentions that 5G scenarios pose challenges to intrusion detection robustness, which is inconsistent with the later discussion of issues like class imbalance.

2. The abstract states that the proposed model integrates FedProx, but the method and experiments use FedAvg.

3. The abstract states that the proposed model integrates knowledge distillation and federated learning to improve performance and reduce computational overhead (no theoretical or experimental verification). However, the purposes of knowledge distillation and federated learning are model lightweighting and privacy protection, respectively. Therefore, they are not causally related to performance improvement. Furthermore, because a teacher model needs to be trained on the server side using all the data, and then distillation and classification training is performed separately for each client, incurring communication overhead, the total overhead actually increases.

4. In the contribution statement, points 1 and 2 are combined, and points 3 and 4 are combined; there is no need to separate them.

5. In Section 3.1, "..., where benign traffic is extensively outnumbered by malicious instances"?

6. Section 3.1 lacks detailed descriptions of sampling.

7. Section 3.2 lacks further detail regarding model architecture and communication. Furthermore, it fails to explain why DistilBERT was chosen, nor does it demonstrate its superiority over other architectures through experiments.

8. Section 3.3 only describes the loss function for the student model, omitting the loss function for the teacher model. When the teacher model is trained, the student model typically only undergoes distillation; why include a classification loss? Additionally, the $\alpha$ setting of 0.5 and the temperature scaling factor of 10 are provided without textual explanation or gridded experiments.

9. Section 3.4 neither provides the aggregation formula nor cites FedAvg's original literature.

10. Sections 3.5, 3.6, 4, and 5.6 are essentially the same thing, occupying a large amount of space but providing little information.

11. In Algorithm 1, the font used for the loss function is much larger than the surrounding text, the "0:" in each line needs to be removed, and the aggregation formula used is inconsistent with FedAvg's.

12. In Section 5.1, the proposed model and experiments only involve the collected dataset; therefore, the data collection process is not the focus and could be placed in the appendix. Given its commonality, it could even be omitted. Furthermore, the dataset used is not attributed or provided for download, making its value incalculable.

13. The experiments lack multi-dimensional comparisons, such as richer datasets, newer contrast algorithms, ablation studies of components (DistilBERT, FedAvg, etc.), and grid testing of parameters.

14. The layout of Figure 4 is aesthetically unappealing.

15. The writing and structure still have significant room for improvement.

16. Although the paper uses 5G intrusion detection as a context, it does not explore the unique problems of this scenario; its essence is still anomaly detection, merely replacing the experimental dataset with a traffic dataset. For anomaly detection, the paper only integrates existing techniques such as balanced sampling, DistilBERT, and FedAvg. Moreover, the entire methodology/framework section only occupies two pages. Combined with points 10 and 13 above, the workload is far from sufficient.

17. Not only are anomalous samples few in number, but their types are also difficult to collect comprehensively. For the former, sampling/generating anomalous samples is reasonable, ensuring both class balance and sufficient training data. However, this paper samples an equal number of normal samples as anomalous samples, leading to a few-shot scenario and causing problems with generalization. For the latter, modeling only the normal class or additionally modeling an unobservable anomalous class would be more realistic, but this paper simply treats anomaly detection as a regular classification task. Furthermore, unsupervised or weakly supervised models are generally more suitable for practical needs and have attracted extensive research, while this paper remains at the fully supervised level.

18. The purpose of federated learning is data decentralization, with clients not sharing data. However, the proposed architecture has a teacher model on the server side, requiring all the data to be fed for training. Additionally, the paper seems to mention a global model $S_{global}$ for validation, rather than validation by each client individually. These factors confuse me regarding the meaning of the federated setup in this paper.

**Questions:**

See Weaknesses.

---

### Official Review · Reviewer_G7zz · 2025-10-31

**Soundness:** 2
**Presentation:** 2
**Contribution:** 2
**Rating:** 2
**Confidence:** 4

**Summary:**

The paper proposes FEDAKD, a federated pipeline for 5G intrusion detection that (i) rebalances data via “anomaly-aware sampling,” (ii) trains a centralized teacher transformer, and (iii) distills to client-side student models, aggregating with a FedAvg-style scheme. On a custom 5G dataset, the method reports very high weighted F1 and large gains over FedAvg/FedProx/FedDyn/FedMD and several centralized baselines.

**Strengths:**

Addresses an important application (privacy-preserving intrusion detection with severe class imbalance).

Teacher–student KD for edge constraints is a sensible engineering choice.

Reported gains on the provided dataset are large.

**Weaknesses:**

Contributions not clearly defined. Listing experiments or general analysis as contributions is not acceptable for ICLR. No concrete, novel technical contribution is isolated.

Not novel vs. FL+KD literature. The method looks like a standard teacher–student overview with no new formula, loss, or mechanism beyond well-known KD in FL.

Anomaly-aware sampling breaks FL assumptions. In FL, clients keep their own data; you cannot centrally rebalance/assign anomalies to clients. Evaluate under proper heterogeneity (e.g., label/quantity skew) without reassigning data.

No principled proof/justification. There is no theoretical or algorithmic justification for the method; claims are not backed by analysis beyond empirical curves.

Baselines look misconfigured. Reporting ~0 accuracy/F1 for FedProx/FedDyn without hyperparameters is a red flag. With certain μ, FedProx ≈ FedAvg; zeros suggest config or implementation issues.

Writing/presentation issues. FedProx is mentioned in the abstract but it's just a baseline; FedProx is repeated twice in contributions; several phrasing/consistency problems.

Low-quality figures and typography. Overview figure is small/low-res; algorithm fonts are inconsistent/buggy; the visualization quality makes understanding harder.

Small-scale experiments. Only 5 clients and few rounds; single dataset; results can be biased. Need more clients/rounds/datasets to claim robustness.

Missing details/ablations/code. Key hyperparameters (e.g., FedProx μ, FedDyn settings, sampling fractions) are absent; no ablation to separate the effects of sampling vs. KD; code not provided, so results cannot be verified.

**Questions:**

1.	Is anomaly-aware sampling a purely centralized preprocessing step? If so, how is that compatible with FL privacy/data-locality? If not, how is it implemented locally without global access to raw data or labels across clients?
	2.	What is the genuine novelty in your KD beyond standard CE + KL with temperature? How does this differ from prior FL+KD (e.g., FedMD and related work)?
	3.	Exact baseline configurations: FedProx μ, FedDyn settings, FedMD protocol, and any class-imbalance mitigation for baselines (e.g., class weighting, focal loss, local oversampling).
	4.	How are performance weights for aggregation computed (on which split), normalized, and stabilized?
	5.	Why add Gaussian noise in aggregation? If for DP, provide (ε, δ) accounting and clipping; if not, justify.
	6.	Results with more clients/rounds and stronger non-IID (including clients with zero samples of some classes). Plans or evidence for scalability?
	7.	Will you release code and the 5G testbed pipeline to enable reproduction?

---

### Official Review · Reviewer_t7oX · 2025-11-01

**Soundness:** 1
**Presentation:** 1
**Contribution:** 2
**Rating:** 2
**Confidence:** 4

**Summary:**

The paper tackles the challenge of maintaining effective intrusion detection in complex and high-volume 5G network environments. The authors propose Federated Edge-Assisted Anomaly-Aware Knowledge Distillation (FEDAKD) framework, which combines anomaly-aware sampling, a teacher–student transformer architecture, and enhanced aggregation methods like FedProx to improve detection performance while reducing computation cost.

However, there is still room for improvement in the experimental and analytical sections. Specifically, the paper would benefit from a more consistent description of baseline methods, as some methods mentioned in the tables are not fully explained in the text. In addition, the lack of computational complexity and communication cost analysis limits the understanding of the method’s efficiency in practical scenarios.

**Strengths:**

The choice of baselines is comprehensive and demonstrates novelty in experimental design.

**Weaknesses:**

-- The paper contains inconsistencies between the description and the experimental table regarding the baseline methods.
In the Results and Discussion section, the authors state that the proposed FEDAKD method is compared with three federated baselines (FedAvg and FedProx) and two centralized baselines (RandomForest and LogisticRegression). However, this description is inconsistent with the experimental table, which lists FedAvg, FedProx, FedDyn, FedMD, LSTM, RandomForest, and LogisticRegression as baseline methods. The text not only omits the mention of FedDyn and FedMD but also incorrectly claims “three federated baselines” while introducing only two. This inconsistency may cause confusion for readers and affects the clarity and credibility of the experimental setup.

-- The paper lacks a comprehensive analysis of computational complexity and communication cost.
In the algorithm design section, there is no discussion or theoretical analysis of the computational complexity of the proposed FEDAKD framework. Similarly, the experimental section does not include any evaluation of computation overhead, communication cost, or scalability with respect to the number of clients. Since these factors are critical for assessing the feasibility and efficiency of federated learning in real 5G environments, the absence of such analysis weakens the completeness and practical value of the study.

-- The paper does not include an ablation study to verify the contribution of each component of the proposed method.
The experimental section lacks an ablation analysis that isolates and evaluates the impact of key components within the FEDAKD framework, such as anomaly-aware sampling, the teacher–student transformer design, and the FedProx-based aggregation strategy. Without such experiments, it is difficult to determine how much each module contributes to the overall performance improvement. Including an ablation study would significantly strengthen the persuasiveness and interpretability of the results.

**Questions:**

Please see weaknesses.

---

### Note · Authors · 2025-12-04

I have read and agree with the venue's withdrawal policy on behalf of myself and my co-authors.